# Children's descriptions of playing and learning as related processes

**Susan M. Letourneau**[1] *, **David M. Sobel**[2]

**1** New York Hall of Science, New York, New York, United States of America, **2** Brown University, Providence, Rhode Island, United States of America

* sletourneau@nysci.org

**Data Availability Statement:** The consent forms provided to participants did not include information about public data sharing. Therefore the data are restricted. Subject to IRB approval, researchers may request access to the data by contacting

## Abstract

Many studies have examined children's understanding of playing and learning as separate concepts, but the ways that children relate playing and learning to one another remain relatively unexplored. The current study asked 5- to 8-year-olds (N = 92) to define playing and learning, and examined whether children defined them as abstract processes or merely as labels for particular types of activities. We also asked children to state whether playing and learning can occur simultaneously, and examined whether they could give examples of playing and learning with attributes either congruent or incongruent with those activities. Older children were more likely to define both playing and learning in terms of abstract processes, rather than by describing particular topics or activities. Children who defined both playing and learning in this way were able to generate more examples of situations where they were simultaneously playing and learning, and were better able to generate examples of learning with characteristics of play, and examples of playing with characteristics of learning. These data suggest that children develop an understanding that learning and playing can coincide. These results are critical to researchers and educators who seek to integrate play and learning, as children's beliefs about these concepts can influence how they reflect on playful learning opportunities.

## Children's developing understanding of the relation between playing and learning

Early childhood education has increasingly focused on play as a foundation for learning, drawing on decades of research linking children's play with their social and cognitive development [1–6]. This work has shown that play provides opportunities for children to practice social and emotional skills, to use increasingly complex cognitive processes, and to strengthen bonds with their caregivers and peers [7–9]. Play can also support more formal learning outcomes, particularly with adult guidance [10–13]. In sum, play is an avenue for many kinds of learning in early childhood.

Despite this evidence, studies have also found that children often describe playing and learning as mutually exclusive. From a young age, children describe play as a freely-chosen and social activity that involves positive affect, while learning is mandatory, serious, and overseen by adults [14–20]. The methods used in many of these studies, however, might encourage

ResearchData@brown.edu, attention Arielle Nitenson.

**Funding:** This research was funded by National Science Foundation 1420548 and 1661068 to DMS. The funders had no role in study design, data collection and analysis, decision to publish, or preparation of the manuscript.

**Competing interests:** The authors declare that they have no competing interests.

children to contrast playing and learning without also providing opportunities for them to describe their similarities. For example, children are often asked to describe how playing and learning differ or to label an activity as either playing or learning in a forced-choice task [15, 21]. By presenting playing and learning as opposites, these methods potentially underestimate the extent to which children recognize that playing can lead to learning or that learning can occur while playing.

In this study, we examined how children reflect on the intersections between playing and learning. In particular, we asked whether children who recognize that learning is an active process also recognize that play offers opportunities to learn, and whether this understanding develops over time. Just as adults' awareness of the learning opportunities in play are vital in fostering playful forms of early learning [10, 22–23], children's own metacognitive awareness of how they think and learn can have powerful implications for their engagement in learning as well as their identities as active learners [24–29]. For educators who provide playful learning environments for young children, understanding how children describe their own play and learning can suggest opportunities to scaffold their reflection about what it means to learn, as well as the ways that learning can happen through everyday experiences like play [24, 30].

Numerous studies that have shown that young children develop the capacity to reflect on their own learning [31–38]. For example, in one study, researchers asked children to define "learning" and to give examples of how they had learned in the past [38]. Four- and 5-year-olds often defined learning as tied to particular types of content or topics (e.g., learning is math). By age 8, almost all the children in their sample described learning as an active process that resulted in a change in knowledge or skills, reflecting a metacognitive understanding of learning as involving their own mental states. Independent of age and language abilities, children's definitions of learning related to their ability to describe sources and strategies that allowed changes in their knowledge to take place. Such development is consistent with other investigations of children's understanding of learning, such as their ability to track how or from whom they learned new information [39, 40] or that learning involves integrating various mental states together, and is not dependent on a single action or mental state [41].

Other studies suggest that articulating an abstract, process-based definition of a concept may be domain-specific. For example, similar shifts from concrete to abstract definitions have been found in children's developing concepts of pretending [42], of teaching [43], and of creativity. Children's descriptions of learning as a process of knowledge change, however, developed earlier than their descriptions of teaching as a process that causes knowledge change in others. The question remains whether children also come to define playing as an abstract, metacognitive process. If children do so, when and how do they begin to reflect on the relations between playing and learning, and is a process-based understanding of learning or playing necessary to integrate these concepts?

We asked children between the ages of 5 and 8 to define both playing and learning. We focused on this age group because the studies described above found that children's definitions of learning changed during this time period, shifting from describing particular topics that could be learned to describing a process through which they learned. By asking children about both playing and learning in the current study, we examined whether children had abstract, process-based understandings of both concepts. Moreover, asking about both concepts allowed us to directly compare the developmental trajectories of children's responses.

We next asked children to think of examples of activities in which they were both playing and learning at the same time. Our hypothesis was that children who defined both playing and learning as more abstract processes would be more likely to generate examples of activities that they considered to be both playing and learning, and to articulate why those activities could be categorized in both ways. This pattern of findings would suggest that children with more

abstract definitions of these concepts have a metacognitive awareness of when the processes of playing and learning can overlap.

Finally, using a between-subjects design, half of the children in the study were asked for examples of playing that involved features congruent with play (instances when playing was fun, freely chosen, or not directed by adults), and examples of learning that involved features congruent with learning (instances when learning was serious, not freely chosen, or directed by adults). The other half of children were asked for examples of playing and learning with qualities of the opposite activity (i.e., examples of playing incongruent with play and examples of learning incongruent with learning, such as learning that was fun, or play that was serious). These examples came from the previous studies that asked children to describe playing and learning using forced-choice methods [15, 21]. If children use these features to differentiate playing and learning, then they should have more difficulty coming up with examples when given incongruous rather than congruous qualities. Moreover, their ability to come up with examples with incongruent features might relate to the ways in which they defined these concepts. An open question is whether children's definitions of playing or learning relate to the inferences they make about whether playing or learning is occurring.

## Methods

### Participants

Participants included 92 children (57 girls, 35 boys) between the ages of 5 and 8 (Range: 60.20–107.90 months, $M$ = 84.96 months). Children were tested at a local children's museum during regular museum visits with a family member or guardian present. No formal measures of race, ethnicity or SES were administered, but the majority of children were white and middle to upper-middle class (as reflected by museum visitor surveys).

### Procedure

This research was approved by the Brown University IRB under the protocol, *Emergence of Diagnostic Reasoning and Scientific Thinking (#1201000538)*. Interviews took place in a quiet room within the museum and lasted approximately 10 minutes. All parents/guardians were stepped through informed consent and children had to agree to participate before the experiment started.

The first part of the procedure involved asking children to define learning and playing. Children were asked to define learning using prompts from a 2015 study by Sobel & Letourneau [38]. The interviewer asked "What does learning mean?" If children did not respond, the question was restated, "What does it mean to learn?" The interviewer also asked, "What do you think 'playing' means?" If children did not understand the question or did not respond, the question was restated, "What does it mean 'to play'?" If children were not sure or did not answer, the interviewer moved on to the next questions. Whether children were asked to define learning or playing first was counterbalanced.

Children were then asked whether they could think of a time that they were playing and learning at the same time (with the order of the words 'playing' and 'learning' in the question counterbalanced across children) and to describe what they were doing. They were then asked "Why was that both playing and learning?" Children were allowed to generate up to three examples.

Next, children were asked to provide examples of their own playing and learning under different conditions. Approximately half of the children in this sample (n = 45) were assigned to the *congruent* condition, in which they were asked to generate examples of playing under characteristic attributes related to playing (being enjoyable, freely chosen, and without adults) and

examples of learning with attributes related to learning (being serious, mandatory, and with adult supervision or direction). Thus, in the congruent condition, children were asked whether they could think of time they were playing and having fun or being happy, doing something that they wanted to do, and when there were no adults supervising. For each, they were given prompts like "what were you doing?" and "tell me more about that," if necessary. For each example, they were asked whether they were learning too and to justify their answer. Similarly, children in the congruent condition were asked whether they could think of a time they were learning and were being serious or concentrating, doing what someone else told them to do, and were with an adult like a teacher. The same prompts were used, and children were asked whether they were also playing in these examples and to justify their answer.

The other children in the sample (n = 47) were assigned to the *incongruent* condition in which they were asked to generate examples of playing with characteristic conditions related to learning, and examples of learning with characteristic conditions related to playing. These children were asked if they could think of a time when they were playing and were serious or concentrating, doing what someone else told them to do, and playing with adult supervision. Similarly, these children were asked if they could think of a time when they were learning and having fun or being happy, doing what they wanted to do, and without adult supervision. The same prompts and follow-up questions were used. The order in which they received the questions about playing and learning were counterbalanced.

## Coding

Children's definitions of learning were categorized in the same manner as Sobel and Letourneau (2015) [38] in order to replicate their findings and analyze the shift from more concrete example-based to more abstract, process-based definitions of learning. Responses were divided into the following categories: (1) *No Response*, including "I don't know," or no answer; (2) *Identity* responses, in which children used the word "learn" or "learning" to define learning (e.g., "learning is when you learn."); (3) *Content* responses, in which children defined learning as involving a subject or topic that was or could be learned (e.g., "Like reading and math."), and (4) *Process* responses, in which children defined learning as involving either a source (e.g., "when your teacher tells you something") or a strategy ("when you practice again and again until you know it") that would result in gaining knowledge.

Definitions of playing were coded into the following categories, in order to distinguish more concrete example-based definitions with more abstract process-based definitions: (1) *No response*, or "I don't know". (2) *Identity*: the child used the word "play" or "playing" to define playing, without elaborating further (e.g., "Playing is when you play."). (3) *Content*: the child's answer contained information about *what* they play or play with (e.g., "Using your toys."). (4) (4) *Process*: the child's answer contained information about either *who* they play with (e.g., "Hanging out with your friends"), *how* they play (e.g., "chasing each other", "building things", "pretending"), or the *outcome or result* of playing (e.g., "having fun", "being happy"). We combined these three aspects of children's definitions of playing because they align with the types of sources and strategies that were included in children's process definitions of learning. With the exception of the no response category, these categories were not mutually exclusive; children could mention more than one aspect of play in their definitions.

We next looked at the examples in which children described themselves as playing and learning at the same time. First, we coded how many examples children were able to generate (ranging from 0 to 3). Next, we coded what children described playing or learning in each example. Coders judged whether children's examples involved one of the following forms of play: *physical play* (e.g., playing tag, sports), *a structured indoor game* (e.g., board games,

puzzles), *creative play* (e.g., drawing, painting), *pretend play* (e.g., playing house), or *functional object play* (playing with toy cars), or were not examples of playing. Coders also judged whether children's examples involved one of the following types of learning: *topics* (such as general academic or protoacademic subjects, like math or colors), *skills* (such as physical skills like learning how to swim or other instructions, like how to make a bracelet), *conventions* (such as social and nonsocial rules like "wear a coat outside" or "it's nice to share"), or *facts* (such as non-generalizable knowledge like "ants have six legs"), or were not examples of learning. These codes were similar to the ones used in our prior study on children's definitions of learning [29], and were meant to document the types of activities that children judged to be both playing and learning. Finally, we coded whether children generated examples of playing and learning in response to each individual attribute (e.g., having fun/being serious, directed/ not directed by an adult, doing what someone tells you to do/doing what you want to do), using a binary code.

Children's definitions of learning and playing were all coded from transcripts of the interviews by two undergraduate research assistants who were both blind to the purpose of the study. Overall agreement was 95% (Kappa = .75). Disagreements were resolved by the first author. The rest of the coding was performed by two different undergraduate research assistants, who were also blind to the purpose of the study. Their agreement was 91% (Kappa = .79). Disagreements were resolved by the second author.

## Statistical analyses

All statistical analyses were conducted using SPSS Statistics software for Windows, Version 24 (IBM Corp., Released 2016). To protect the privacy and confidentiality of participants in this study, only de-identified data will be made available to interested researchers. These data are located at https://doi.org/10.26300/gtrw-7q13 through the Brown University Data Repository System. Data sharing is contingent on IRB approval from the requester's home institution.

We conducted our analyses as follows. First, to determine how children's definitions of playing and learning changed with age:

1). We determined whether children generated more abstract, metacognitive definitions of playing and/or learning This included *process-based* definitions of learning (in which children mentioned with whom or how learning occurred) and of playing (in which children mentioned how, with whom, and the results of playing).

2). We calculated correlations between children's metacognitive definitions of playing and of learning with age, and examined the frequency with which children generated metacognitive definitions of either concept. We also calculated partial correlations between these variables controlling for the mean length of utterances in children's definitions of playing and of learning (MLU).

Next, to understand how children believed that playing and learning related to one another:

3). We examined the number of examples of activities that children considered to be both playing and learning at the same time, and calculated correlations among this variable, children's age, and the presence of metacognitive definitions of playing and of learning. We also qualitatively described the types of examples children gave.

4). We conducted a multinomial logistic regression to determine the unique contributions of children's definitions of playing, of learning, and age on the number of examples they gave of playing and learning at the same time.

5).   We examined children's ability to generate examples of playing and learning in the congruent vs. incongruent condition. We calculated the total number of examples children generated; children could generate up to three examples of playing and up to three examples of learning, since children answered three questions about the characteristics of each activity. We used a General Estimating Equation Analysis, analyzing the total number of examples of each type that children generated in an ordinal logistic model, with play vs. learning as a within-subject factor, condition and whether children generated metacognitive definitions of learning and play as between-subject factors, and age (in months) as a covariate. This analysis shows whether children had difficulty generating examples of playing with characteristics of learning, and vice versa.

6).   Finally, we examined each characteristic individually as they related to children's judgments of playing and learning. We used Fisher's exact tests to determine whether there were differences in children's likelihood of generating an example for playing vs. learning for any individual characteristic (e.g., how often children generated an example of having fun while playing vs. while learning), and Chi-Squared tests to determine whether there were differences between each congruous and incongruous characteristic (e.g., generating an example of playing while having fun vs. while being serious).

We also note that although we used a task that relied on children's linguistic responses, we controlled for MLU in our analyses of children's definitions (see Results), and our other analyses focused on whether children generated any valid response, and not the amount of detail or length of their responses. For example, when asked if they could think of a time when they were playing and learning at the same time, children's answers could be extremely brief ("Yes, hopscotch") and still be considered valid because they show that children themselves thought this activity involved some aspect of playing and some aspect of learning. We did ask children to justify their answers in order to prompt them for as much detail as possible to aid in coding, but our analyses were based on the presence of particular responses to our questions, rather than their length. Therefore, we believe this linguistic task is an appropriate method for querying children's conceptions about what it means to be playing or learning, as our primary concern was making the task as open-ended as possible to avoid presenting playing and learning as opposites.

## Results

### How did children's definitions of playing and learning change with age?

Table 1 shows the distribution of children's definitions of playing and learning. Our first analyses focus on whether children generated metacognitive (i.e., process) definitions of playing

**Table 1.  Distribution of children's definitions of playing and learning.**

| Response Type | Playing | | Learning | |
|---|---|---|---|---|
| Playing | N | % | N | % |
| No response | 3 | 3.26 | 8 | 8.70 |
| Identity | 7 | 7.61 | 9 | 9.78 |
| Content | 30 | 32.61 | 26 | 28.26 |
| Process | 74 | 80.22 | 61 | 66.30 |

With the exception of "No response," codes are not mutually exclusive, so percentages can add up to more than 100%.

and learning. There were no differences in these definitions between genders, $\chi^2(1, N = 92) =$ 0.21 and 0.13 for playing and learning respectively, $p = .65$ and .72, so this variable will not be considered further. We examined how age and MLU correlated with metacognitive process definitions of learning and of playing. There were positive correlations between children's age and MLU for their definitions of learning, $r_s(90) = .30$, $p = .003$, and their definitions of playing $r_s(90) = .17$, $p = .11$. MLU values significantly correlated with the presence of metacognitive process definitions of learning, $r_s(90) = .42$, $p < .001$, and of playing $r_s(90) = .20$, $p = .05$. We observed a significant positive correlation between age and metacognitive process definitions of learning, $r_s(90) = .40$, $p < .001$. Partial correlations showed that this effect was still significant after controlling for the MLU in children's definitions, $r_s(87) = .32$, $p = .002$. These findings paralleled the results of Sobel and Letourneau (2015) [38]. There was also a significant positive correlation between age and metacognitive definitions of playing, $r_s(90) = .34$, $p = .001$, and again, this correlation remained significant when controlling for the MLU of children's definitions, $r_s(87) = .31$, $p = .003$. Unsurprisingly, there was also a significant correlation between children's age and whether their definitions of both learning and playing were coded as metacognitive, $r_s(90) = .39$, $p < .001$. Fig 1 shows the relation between children's age and whether they generated a metacognitive definition of learning and playing.

We compared the frequency with which children generated metacognitive definitions of learning versus playing. Overall, children were more likely to generate metacognitive definitions of playing than learning, McNemar $\chi^2(1, N = 92) = 6.26$, $p = .01$. Fifty-six children (60.87%) generated abstract metacognitive definitions of both concepts, and 18 children (19.57%) generated such a definition of play but not learning, while only 4 (4.35%) generated such a definition of learning but not play, and 14 (15.22%) generated no such definitions.

## How did children believe that playing and learning related to one another?

To answer this question, we first examined the number of examples children gave of playing and learning together. The frequency of such examples is shown in Table 2. The number of

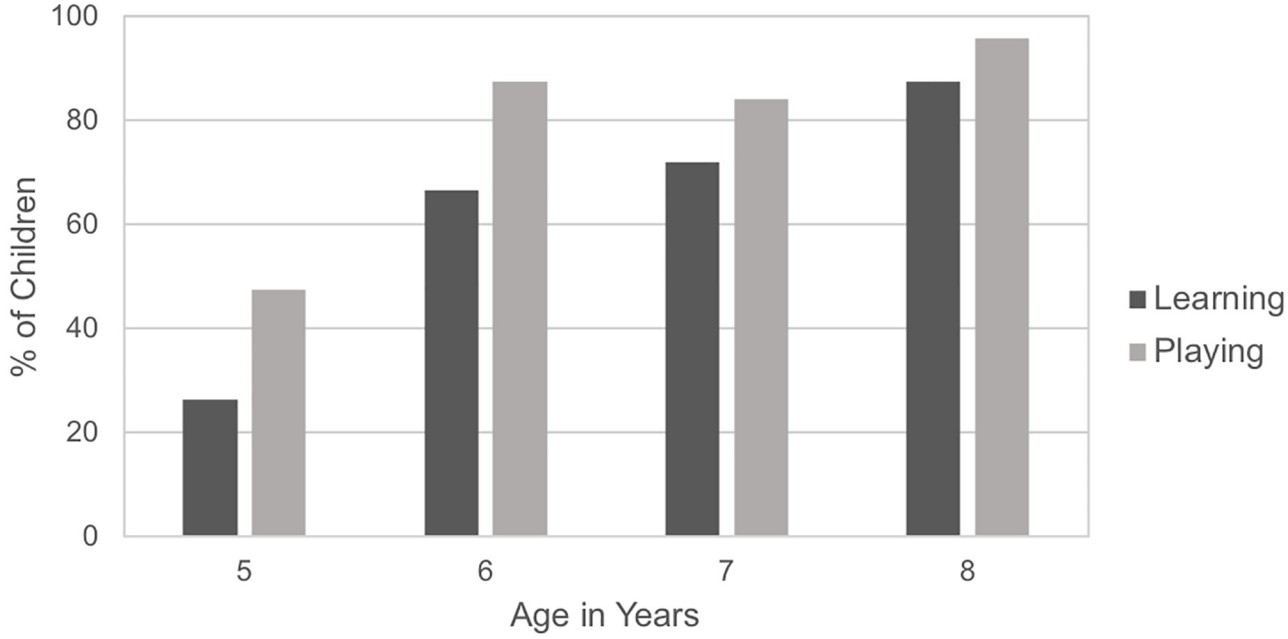

**Fig 1. Percentage of children providing metacognitive definitions of learning and of playing, by age.**

**Table 2. Number of children generating at least one example of each type of activity coded as playing and learning (excluding invalid cases).**

| Learning Code | Play code | | | | |
| --- | --- | --- | --- | --- | --- |
| | Physical Play | Indoor Games | Creative Play | Pretend Play | Functional Object Play |
| Topic | 4 | 25 | 9 | 2 | 0 |
| Skill | 14 | 5 | 4 | 1 | 0 |
| Convention | 1 | 0 | 0 | 1 | 1 |
| Fact | 1 | 5 | 4 | 0 | 0 |

examples children generated correlated with age, $r_s(90) = .38$, $p < .001$, as well as with the presence of abstract metacognitive definitions of learning, $r_s(90) = .37$, $p < .001$, and playing, $r_s(90) = .33$, $p = .001$. The number of examples that children generated was also correlated with the presence of such definitions of *both* play and learning, $r_s(90) = .38$, $p < .001$, and this correlation held when controlling for age, $r_s(89) = .27$, $p = .01$.

To isolate the specific contribution of these predictors, we ran a multinomial logistic regression on the number of examples children generated. This showed an overall significant model, $\chi^2(9) = 28.08$, $p = .001$. There was no unique effect of age, -2 log likelihood = 208.55, $\chi^2(3) = 3.79$, $p = .29$, nor a unique effect of whether children generated a metacognitive aspect of playing in their definition, -2 log likelihood = 210.20, $\chi^2(3) = 5.44$, $p = .14$. There was a unique effect of whether children generated an abstract metacognitive definition of learning, -2 log likelihood = 212.66, $\chi^2(3) = 7.91$, $p = .05$.

Table 2 also shows the types of examples of playing and learning that children generated. When children generated examples of playing and learning together, they fit into one of three categories: Children talked about engaging in physical activities that allowed them to learn particular skills relevant to that activity (e.g., playing on the monkey bars allowed them to learn how to climb on the bars), engaging in structured indoor activities that involved particular topics (such as playing math games), and engaging in creative activities that allowed them to learn topics (such as drawing and learning about letters). Whether children generated at least one of these examples correlated with whether they generated process-based definitions of *both* play and learning, $r_s(90) = .33$, $p = .001$, and this correlation held when controlling for age, $r_s(89) = .26$, $p = .01$.

We then examined the number of examples children generated in the congruent versus incongruent condition. Recall that children were asked whether they could think of a time when they learned with particular attributes related to learning (congruent condition) or playing (incongruent) and playing with attributes related to playing (congruent condition) or learning (incongruent condition). We found a unique effect of condition, with children generating more examples in the congruent than the incongruent condition, Wald $\chi^2(1) = 7.33$, $p = .007$, as well as a unique effect of generating a metacognitive definition of learning, Wald $\chi^2(1) = 6.48$, $p = .01$. The unique effect of generating a metacognitive definition of playing was marginally significant, Wald $\chi^2(1) = 2.93$, $p = .09$. Age did not uniquely predict variance in this model, Wald $\chi^2(1) = 1.04$, $p = .31$.

Table 3 shows the frequency with which children generated a valid example for each question. As confirmed by the analysis above, children always generated more examples of playing and learning when presented with congruent rather than incongruent attributes. When each attribute was analyzed individually, only one difference reached significance: children generated more examples of playing while having fun than learning while having fun, Fisher's Exact Test, $p = .001$. Responses to playing vs. learning with no adults, learning vs. and playing with adults, and learning vs. playing while being serious were all marginally significant, Fisher Exact Tests, $p = .10$, $.06$, and $.07$ respectively.

**Table 3. Proportion of children who generated a valid example of play or learning (in parentheses) based on condition.**

|  | Doing what you want | No Adults | Having Fun | Someone told you | With adult | Being Serious |
|---|---|---|---|---|---|---|
| Congruent Condition | (Play) 69 (.47) | (Play) .55 (.50) | (Play).96 (.21) | (Learning) 51 (.51) | (Learning) .84 (.37) | (Learning) 67 (.48) |
| Incongruent Condition | (Learning).64 (.49) | (Learning) 40 (.49) | (Learning) 70 (.46) | (Play) 49 (.50) | (Play) 68 (.47) | (Play) 49 (.51) |

Top parentheses show which question was asked. In the congruent condition, children were asked to provide examples of times they were playing and doing what they wanted, with no adults, and having fun and examples of times they were learning when someone told them what to do, with an adult, and while being serious. In the incongruent condition, they were asked about play when someone told them what to do, with an adult, and while being serious and learning while doing what they wanted, with no adults, and while having fun. Bottom parentheses shows standard deviation.

When we compared congruous versus incongruous characteristics individually, children were also more likely to generate examples of playing while having fun than while being serious, $\chi^2(1, N = 92) = 24.64$, $p < .001$, Phi = .52, and when choosing what to do than being told, $\chi^2(1, N = 92) = 3.78$, $p = .05$, Phi = .20. When we conducted the same contrasts for learning, and children were more likely to generate example of learning with an adult than without, $\chi^2(1, N = 92) = 18.90$, $p < .001$, Phi = .45.

Definitions of playing and learning had little relation to children's examples of playing and learning in the congruent condition after controlling for age. Children with metacognitive definitions of both play and learning were more likely to generate an example of learning when someone told them what to do, $r_s(43) = .33$, $p = .03$, but this correlation was not significant when age (in months) was controlled for, $r_s(42) = .21$, $p = .17$. No other attributes correlated with children's definitions of playing or learning in the same condition. In contrast, in the incongruent condition, children who generated metacognitive definitions of both concepts were more likely to generate examples of play and learning with characteristics of the opposite activity—including learning while having fun, $r_s(44) = .44$, $p = .002$, playing when someone told you what to do, $r_s(44) = .41$, $p = .005$, and playing with an adult, $r_s(44) = .35$, $p = .02$). All of these effects remained significant ($p \leq .05$) when controlling for age.

## Discussion

The present study used structured interviews to examine children's explicit understanding of the meaning of playing and learning, and the relation between the two concepts. We found that children articulate an understanding of playing and learning as abstract processes that can happen simultaneously and share characteristics. When asked to define learning and playing, younger children in our sample were frequently unable to offer any definition, and when they did so, they focused on content (what they played or what objects they played with). In contrast, the older children in our sample were more likely to define playing based on how they played or the result of their playing. The results on learning replicate our prior findings [38], and more generally, they suggest a developmental shift toward describing both playing and learning as processes with distinct outcomes rather than using these words as labels for certain types of activities.

Articulating abstract definitions of playing developed earlier than similar articulations of definitions of learning. We speculate that children might initially have separate concepts of playing and learning. With a more sophisticated understanding of the processes involved in both playing and learning, children may develop a more undifferentiated concept that learning and playing can co-occur, depending on the qualities of a given activity. Further, children's understanding of learning as a metacognitive process might function as a bottleneck in their ability to see play and learning as related. Children who generated abstract definitions of both concepts were more likely to generate examples of activities they considered to be both playing

and learning, but it was whether children defined learning as an abstract process that was predictive. Importantly, many of the findings held when controlling for age, suggesting that other developing factors like cognitive or language capacities were not solely responsible for the development we observed.

Children who articulated abstract definitions of playing and learning were also better able to describe examples of playing with qualities of learning, and vice versa. That said, children did generate more examples of learning and playing when given congruent than incongruent attributes, suggesting that they believe certain qualities are more characteristic of one activity or the other. Children were also more likely to state that their examples of play were also examples of learning (regardless of whether the attributes inherent in the activity related to learning) than to state that their examples of learning were also play. This is also consistent with the hypothesis that children's understanding of learning as a metacognitive process might be critical for realizing that playing and learning can be related to one another. Knowing that learning is an abstract process (as evidenced by their definition of learning) might allow children to recognize that activities like playing offer the opportunity to learn. By asking children not only for open-ended definitions of playing and learning, but also for specific examples, this study provides a more detailed description of children's understanding of the overlap between playing and learning; their open-ended definitions reveal a belief that playing and learning are potentially related, and their examples show qualities that make playing and learning both compatible and distinct. Given that adults do not always recognize the learning opportunities in play [22], these findings show that children may be more flexible in their perceptions of the overlap between play and learning.

These interviews show that children are not only capable of reflecting on their learning, but also of reflecting on how learning can occur through play. In addition, the findings suggest that this ability is not solely dependent on age, but is tied to children's conceptual understanding of what it means to learn. An open question is how children's perceptions and attitudes are shaped by their early experiences. What experiences support children's understanding of learning as an active process, and their reflection about learning that might occur in their own play? Do these types of experiences foster a metacognitive understanding of both concepts and allow children to recognize the overlap between playing and learning at younger ages? Moreover, caregivers' and teachers' views about play and learning, and the interactions and educational practices that stem from these beliefs, may also impact children's exposure to and interpretation of playful learning experiences in everyday life [22, 30].

Finally, recognizing how young children understand the intersections between playing and learning has implications for formal and informal education. For example, many informal learning environments use playful approaches to encourage and support learning, but the efficacy of such approaches might be dependent on children's belief that learning can occur during play [24], and the opportunities they receive to reflect on playing and learning together, rather than separately. Children's definitions of learning were most predictive in this study, and previous studies have shown that children are able to reflect on their own learning with prompting. Although we did not gather information about the types of schools that children attended in this study, future studies might examine the impact of different educational approaches and pedagogical strategies on children's perspectives about play and learning. Educators may be able to scaffold children in reflecting on specific instances when they have learned while playing, supporting their metacognitive understanding of the many ways that learning can take place. Developing a metacognitive understanding of learning, and recognizing that learning occurs through everyday experiences like play, may also affect children's overall engagement in learning and conceptions of themselves as learners [24–30]. A next step in this investigation is to see whether children's beliefs about learning, including their self-

efficacy and motivation to learn, is related to the way they play, and in turn, whether valuing and engaging in play can affect their identity as active learners.

## Acknowledgments

We thank Chris Erb, Deanna Macris, and Tiffany Tassin for helpful discussion and Charlotte Crider, Rose DeRienzo, Julia Donovan, Isobel Heck, Colton Lacy, and Zoe Finiasz for assistance with data collection and analysis. We also thank the families at Providence Children's Museum who participated in this research. Address correspondence concerning this article to: D. Sobel, CLPS Department, Box 1821, Brown University, Providence, RI 02912. Phone: 401-863-3038. Fax: 401-863-2255. Email: Dave_Sobel@Brown.edu.

## Author Contributions

**Conceptualization:** Susan M. Letourneau, David M. Sobel.

**Data curation:** Susan M. Letourneau.

**Formal analysis:** Susan M. Letourneau, David M. Sobel.

**Funding acquisition:** David M. Sobel.

**Methodology:** David M. Sobel.

**Project administration:** Susan M. Letourneau, David M. Sobel.

**Writing – original draft:** Susan M. Letourneau, David M. Sobel.

**Writing – review & editing:** Susan M. Letourneau, David M. Sobel.

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
