## [Decision Letter · Decision Letter 0]

17 Jan 2020

PONE-D-19-32302

Children’s descriptions of playing and learning as related processes

PLOS ONE

Dear Dr. Sobel,

Thank you for submitting your manuscript to PLOS ONE. After careful consideration, we feel that it has merit but does not fully meet PLOS ONE’s publication criteria as it currently stands. Therefore, we invite you to submit a revised version of the manuscript that addresses the points raised during the review process.

First of all please accept my apologies for the delay on this decision. I have now received reviews from two expert reviewers. As you'll see, both reviewers are positively inclined towards this manuscript, and believe that it has the potential to make an important contribution to the field, as do I. I am generally in agreement with the reviewers' critiques, and most of their concerns are relatively minor. That said, each reviewer raises a more serious concern about the paper, both of which I shared. I want to highlight those concerns as particularly important to address if you revise this manuscript. 

Reviewer 1 is troubled by the linguistic nature of the task, and is worried that the conclusion that children show "immature" understanding of playing and learning as processes may reflect task demands rather than a true lack of understanding. Reviewer 1 suggests that you might conduct a similar study using a non-linguistic task. I do not believe this is strictly necessary, but I would like to see a careful argument backing up the construct validity of the linguistic task, as well as a discussion of its limitations with respect to your conclusion. 

Reviewer 2 suggests that the overarching question you are asking could be more strongly motivated. They note that some of this currently appears in the discussion, but should be highlighted in the introduction as well. In my own reading of the paper I do see the theoretical motivation for this work, but I agree that it could be more strongly laid out in the introduction.

I do hope you choose to revise this manuscript, as I think both these issues are addressable and can help clarify and solidify the contribution of this work. Please also be sure to attend to all of the more minor concerns each reviewer raises.

We would appreciate receiving your revised manuscript by Mar 02 2020 11:59PM. To enhance the reproducibility of your results, we recommend that if applicable you deposit your laboratory protocols in protocols.io, where a protocol can be assigned its own identifier (DOI) such that it can be cited independently in the future. For instructions see: http://journals.plos.org/plosone/s/submission-guidelines#loc-laboratory-protocols

We look forward to receiving your revised manuscript.

Kind regards,

Lucas Payne Butler

Academic Editor

PLOS ONE

Journal Requirements:

"This work was supported by the National Science Foundation (1223777 and 1420548 to DMS)."

"No. The funders had no role in study design, data collection and analysis, decision to publish, or preparation of the manuscript."

Reviewers' comments:

Reviewer's Responses to Questions

**Comments to the Author**

1. Is the manuscript technically sound, and do the data support the conclusions?

Reviewer #1: Yes

Reviewer #2: Yes

2. Has the statistical analysis been performed appropriately and rigorously? 

Reviewer #1: Yes

Reviewer #2: Yes

3. Have the authors made all data underlying the findings in their manuscript fully available?

Reviewer #1: No

Reviewer #2: Yes

4. Is the manuscript presented in an intelligible fashion and written in standard English?

Reviewer #1: Yes

Reviewer #2: Yes

5. Review Comments to the Author

Reviewer #1: This manuscript explored children’s conceptual understanding of the relationship between playing and learning between the ages of 5-8-years-old. Children’s conceptual knowledge of playing and learning increased with age: they were more likely to define both terms as active processes and were also better able to generate examples of activities involving both playing and learning with age. Controlling for age, children with a more abstract metacognitive concept of learning were more likely to generate examples of activities involving playing and learning together. Children with a mature metacognitive understanding of playing and learning also were more likely to generate examples of playing with qualities of learning and vice versa. Thus, children’s understanding that play can lead to learning develops with age. Yet irrespective of age, conceptual knowledge of what it means to learn relates to children’s broader understanding of the relationship between play and learning.

Early childhood education has recently focused on play-based learning, yet past work suggests that young children think of play and learning as mutually exclusive. This paper adds to this literature by showing that children’s understanding of the relationship between play and learning develops with age. This work is a first step in a line of research identifying how children’s theories and beliefs about play and learning relate to their actual learning.

The main weakness of this paper is its use of a linguistic task to test conceptual change across development. The use of a linguistic task makes it unclear whether young children’s “immature” metacognitive understanding of learning and play stems from a true lack of conceptual knowledge or constraints on their expressive language ability. Thus, this paper would benefit from using a non-linguistic task, better arguments for their use of a linguistic task and its validity in young children, or stating this as a limitation in the discussion.

If the authors are able to address this major concern (as well as the minor ones laid out below), I believe this manuscript would be suitable for publication at Plos One and add valuable information to understanding the development of children’s conceptual knowledge of the relationship between playing and learning.

Abstract

• The authors should state their developmental change findings in the abstract.

Introduction

• Can the authors better define what would count as evidence for an abstract understanding of learning and playing? (last paragraph page 4).

• Can the authors expand on their hypothesis that children who describe learning and playing as active processes might better be able to generate activities classified as both? How is this evidence specifically for metacognition? Couldn’t children also be using a simple heuristic that active processes share properties without really understanding how play and learning relate?

• Is there any evidence that representing play as learning relates to real world behavior in children?

Method

• How was the sample size chosen? Do the authors have enough power to detect results?

• I found it difficult to follow the analyses in the results section. The authors should add an analysis plan section to the methods to better guide the reader through the results. In the analysis plan, they should state any composite measures they will look at and which models they used and why.

Results

• Why (and how) did the authors combine source, process, and outcome into a single factor? This is a bit strange since it means that process was looked at twice – within this factor and alone. It’s also confusing which outcome measures they looked at individually – I suggest the authors restate them all either here or in the suggested analysis plan section.

• I have some concern about the analyses in Table 1. Since the authors are predicting binary outcome measures, did they use logistic regressions? The authors should also correct for multiple comparisons since all the tests look at features of the same construct (within play or learning). Finally, the small sample sizes in some tests (e.g. No response in playing has an n = 3) render them underpowered. The authors may want to consider a different analysis approach or note this as a limitation.

• A graph of age effects would help the reader understand when kids start verbally endorsing a metacognitive/ process use of play and learning

• Does overall length of verbal responses increase with age?

Discussion

• The authors write that “children were also more likely to state that their examples of play were also examples of learning (regardless of whether the attributes inherent in the activity related to learning) than to state that their examples of learning were also play.” What do adults do? It seems that adults would also be more likely to endorse that play leads to learning than vice versa. Is this a general bias we have or truth?

Reviewer #2: PONE-D-19-32302

Title: Children’s descriptions of playing and learning as related processes

The present study examined 5- to 8-year-olds responses to their definitions of playing and learning, and examined whether children defined them as active processes or by particular types of activities. This is an interesting study and the manuscript is well written.

The introduction provides a nice overview of the literature on children’s views of play, however, the authors should provide a stronger argument as to why they believe the research questions are important. That is, why is it important that we understand children’s perspectives on the distinction between play and learning? How is it relevant for their development or learning, or how children engage in a classroom? The authors do address the importance of the research questions in the discussion, but it is important for the authors to also discuss this in the introduction.

It would be interesting to note what type of schooling the children attended (public schools, Montessori, etc.). It seems like this could influence children’s perspectives.

Were gender and age stratified in the randomization of the two conditions?

The authors should indicate the approximate length of the interviews with the children.

The analyses seem appropriate and the results section is well written.

Again, the discussion indicates why understanding children’s perspectives on play and learning is important to understand. It seems as though it would be better to include this in the introduction as well.

6. PLOS authors have the option to publish the peer review history of their article (what does this mean?). If published, this will include your full peer review and any attached files.

Reviewer #1: No

Reviewer #2: No

---

## [Author Response · Author response to Decision Letter 0]

7 Feb 2020

We have included a document called "Response to Reviewers." This details the response to all reviewer comments.

---

## [Decision Letter · Decision Letter 1]

4 Mar 2020

Children’s descriptions of playing and learning as related processes

PONE-D-19-32302R1

Dear Dr. Sobel,

We are pleased to inform you that your manuscript has been judged scientifically suitable for publication and will be formally accepted for publication once it complies with all outstanding technical requirements.

With kind regards,

Lucas Payne Butler

Academic Editor

PLOS ONE

Additional Editor Comments (optional):

Reviewers' comments:

Reviewer's Responses to Questions

**Comments to the Author**

1. If the authors have adequately addressed your comments raised in a previous round of review and you feel that this manuscript is now acceptable for publication, you may indicate that here to bypass the “Comments to the Author” section, enter your conflict of interest statement in the “Confidential to Editor” section, and submit your "Accept" recommendation.

Reviewer #1: All comments have been addressed

2. Is the manuscript technically sound, and do the data support the conclusions?

Reviewer #1: Yes

3. Has the statistical analysis been performed appropriately and rigorously? 

Reviewer #1: Yes

4. Have the authors made all data underlying the findings in their manuscript fully available?

Reviewer #1: Yes

5. Is the manuscript presented in an intelligible fashion and written in standard English?

Reviewer #1: Yes

6. Review Comments to the Author

Reviewer #1: (No Response)

7. PLOS authors have the option to publish the peer review history of their article (what does this mean?). If published, this will include your full peer review and any attached files.

Reviewer #1: No

---

## [Editor Report · Acceptance letter]

2 Apr 2020

PONE-D-19-32302R1 

Children’s descriptions of playing and learning as related processes 

Dear Dr. Sobel:

I am pleased to inform you that your manuscript has been deemed suitable for publication in PLOS ONE. Congratulations! Your manuscript is now with our production department. 

With kind regards,

on behalf of

Dr. Lucas Payne Butler 

Academic Editor

PLOS ONE